# The Action Potential Clamp Technique as a Tool for Risk Stratification of Sinus Bradycardia Due to Loss-of-Function Mutations in HCN4: An In Silico Exploration Based on In Vitro and In Vivo Data

**DOI:** 10.3390/biomedicines11092447

**Published:** 2023-09-02

**Authors:** Arie O. Verkerk, Ronald Wilders

**Affiliations:** 1Department of Medical Biology, Amsterdam Cardiovascular Sciences, Amsterdam UMC, University of Amsterdam, 1105 AZ Amsterdam, The Netherlands; a.o.verkerk@amsterdamumc.nl; 2Department of Experimental Cardiology, Heart Center, Amsterdam Cardiovascular Sciences, Amsterdam UMC, University of Amsterdam, 1105 AZ Amsterdam, The Netherlands

**Keywords:** sinoatrial node, human, pacemaker activity, hyperpolarization-activated current, *HCN4* channels, cellular electrophysiology, action potential, patch clamp, computer simulations

## Abstract

These days, in vitro functional analysis of gene variants is becoming increasingly important for risk stratification of cardiac ion channelopathies. So far, such risk stratification has been applied to *SCN5A*, *KCNQ1*, and *KCNH2* gene variants associated with Brugada syndrome and long QT syndrome types 1 and 2, respectively, but risk stratification of *HCN4* gene variants related to sick sinus syndrome has not yet been performed. *HCN4* is the gene responsible for the hyperpolarization-activated ‘funny’ current I_f_, which is an important modulator of the spontaneous diastolic depolarization underlying the sinus node pacemaker activity. In the present study, we carried out a risk classification assay on those loss-of-function mutations in *HCN4* for which in vivo as well as in vitro data have been published. We used the in vitro data to compute the charge carried by I_f_ (Q_f_) during the diastolic depolarization phase of a prerecorded human sinus node action potential waveform and assessed the extent to which this Q_f_ predicts (1) the beating rate of the comprehensive Fabbri–Severi model of a human sinus node cell with mutation-induced changes in I_f_ and (2) the heart rate observed in patients carrying the associated mutation in *HCN4*. The beating rate of the model cell showed a very strong correlation with Q_f_ from the simulated action potential clamp experiments (R^2^ = 0.95 under vagal tone). The clinically observed minimum or resting heart rates showed a strong correlation with Q_f_ (R^2^ = 0.73 and R^2^ = 0.71, respectively). While a translational perspective remains to be seen, we conclude that action potential clamp on transfected cells, without the need for further voltage clamp experiments and data analysis to determine individual biophysical parameters of I_f_, is a promising tool for risk stratification of sinus bradycardia due to loss-of-function mutations in *HCN4*. In combination with an I_f_ blocker, this tool may also prove useful when applied to human-induced pluripotent stem cell-derived cardiomyocytes (hiPSC-CMs) obtained from mutation carriers and non-carriers.

## 1. Introduction

Over the past decade, risk stratification has become common practice for a large variety of diseases [1,2,3,4,5], including cardiac rhythm abnormalities [6,7,8,9,10,11,12]. Attempts at arrhythmic risk stratification are frequently based on patient clinical parameters, including electrical history and basic and advanced electrocardiographic indices [9,13,14,15,16,17,18,19,20]. However, the stratification of arrhythmic risk in patients can be difficult and controversial [15,17,21,22,23]. These days, genetic testing for the presence of gene variants is increasingly becoming part of the clinical management and risk stratification of cardiac ion channelopathies [6,15,24,25,26]. Variants of genes underlying the major ion channels involved in cardiac depolarization and repolarization can be classified as ‘pathogenic’, ‘likely pathogenic’, ‘variant of uncertain significance’, ‘likely benign’, and ‘benign’, according to a classification scheme developed by the American College of Medical Genetics and Genomics (ACMG) [27]. The ACMG criteria for the classification of pathogenic and benign variants include in vitro functional data, which for arrhythmia syndromes are mainly based on patch clamp experiments.

The patch clamp technique is a >40-year-old electrophysiological tool [28] and is considered the gold standard in electrophysiology because of its ability to measure both action potentials (APs) and specific membrane currents, and even single-channel currents, in detail [29]. Functional consequences of variations in genes encoding cardiac ion channels (for reviews, see Marbán [30], Schwartz et al. [10], and Wilde et al. [15]) can be studied in a large variety of models, such as isolated cells from genetically modified animals [31,32,33] and human-induced pluripotent stem cell-derived cardiomyocytes (hiPSC-CMs) [34,35,36]. However, mammalian cell lines in which the wild-type and the variant cDNA of a given ion channel can be expressed are still widely used to study the functional properties of ion currents [37,38]. Initially, patch clamp experiments were performed manually, but in recent years, the pathogenicity of *SCN5A*, *KCNQ1*, and *KCNH2* variants involved in the inherited cardiac channelopathies Brugada syndrome and long QT syndrome types 1 and 2, respectively, have also been determined using high-throughput reclassification assays based on automated patch clamp devices [25,39,40,41,42,43,44]. Various biophysical parameters of the membrane currents have been used to classify the risk of genetic variants, but current density is the most reliable marker for predicting the risk of *SCN5A* [45], *KCNQ1* [39], and *KCNH2* [46] ion channel variants, followed by their voltage dependence of activation (through their half-activation voltage, V_½_).

Here, we performed a risk classification assay for sick sinus syndrome, a group of disorders with the common feature that the heart cannot adequately perform its normal pacemaking function [47,48,49]. Impaired cardiac pacemaker function may be due to reduced impulse formation in the sinoatrial node (SAN), defects in impulse conduction from SAN to atria, or widespread atrial electrophysiological abnormalities [49,50,51,52], but in the present study, we limited ourselves to familial sinus bradycardia due to loss-of-function mutations in *HCN4*. The *HCN4* gene encodes the *HCN4* protein, which is the major HCN isoform of the ion channels in the human SAN that mediate the hyperpolarization-activated ‘funny’ current I_f_ (also called ‘pacemaker current’) [53,54]. I_f_ is a depolarizing inward current during the diastolic depolarization phase of human SAN APs and fulfills an important modulatory role [55]. To date, several *HCN4* variants have been identified, as reviewed by us and others [56,57,58,59,60], and they all affect pacemaker function by one or more alterations in the unique set of biophysical parameters of I_f_, including activation upon hyperpolarization, time constants of activation and deactivation, reversal potential, modulation by cAMP, and density of channels. In the present study, we tested whether the affected biophysical parameters of *HCN4* variants can explain the severity of sinus bradycardia. Therefore, we first selected from the literature those loss-of-function mutations in *HCN4* that have been associated with familial sinus bradycardia and for which both clinical and in vitro data are available, with the condition that the clinical data include quantitative heart rate data from at least two mutation carriers. The in vitro data were then used to compute the charge carried by I_f_ (Q_f_) during the diastolic depolarization of a prerecorded human SAN AP waveform [61] as a measure of the physiological impact of I_f_ that can be readily determined in AP clamp experiments without the need for further voltage clamp experiments and data analysis to determine individual biophysical parameters of I_f_ [55,60,61]. We assessed the extent to which this Q_f_ predicts (1) the beating rate of the comprehensive Fabbri–Severi model of a human SAN pacemaker cell [62] with mutation-induced changes in I_f_, and (2) the heart rate observed in patients carrying the associated mutation in *HCN4*. We demonstrate that the beating rate of the model cell, as well as the clinically observed minimum or resting heart rate, show a strong correlation with Q_f_ and conclude that risk stratification by AP clamp is a promising tool for risk stratification of sinus bradycardia due to loss-of-function mutations in *HCN4*.

## 2. Materials and Methods

### 2.1. Simulations of Action Potential Clamp Experiments

A prerecorded AP waveform from a single isolated human SAN cell [61], with a cycle length of 813 ms, was used to construct a train of 100 APs that could be employed as a command signal of ≈82 s duration under voltage clamp conditions, which was long enough to achieve stable behavior of the *HCN4* current during the simulated AP clamp experiments. This typical AP waveform had been recorded from a pacemaker cell isolated from a human SAN in the whole cell configuration of the patch clamp technique [61]. Recordings were made at 36 ± 0.2 °C, and the extracellular Na^+^, K^+^, and Ca^2+^ concentrations amounted to 140, 5.4, and 1.8 mmol/L, respectively, whereas the recording pipette solution contained 145 mmol/L K^+^ and 5.0 mmol/L Na^+^. Of note, these ion concentrations are identical or almost identical to those in the Fabbri–Severi human SAN cell model [62] (Section 2.2). Only the intracellular K^+^ concentration of the model cell (140 mmol/L) is slightly different from the K^+^ concentration in the pipette solution (145 mmol/L).

The custom software to simulate such AP clamp experiments was compiled as a 32-bit Windows application using Intel Visual Fortran Composer XE 2013 and run on an Intel Core i7 processor-based workstation. For the numerical reconstruction of the *HCN4* current, we used equations based on our experimental data on I_f_ acquired from the same set of single isolated human SAN cells [61], as described in detail by Verkerk et al. [63] and also employed by Fabbri et al. [62] in their Fabbri–Severi model of a human SAN pacemaker cell. We applied a simple and efficient Euler-type integration scheme with a time step of 10 μs for the numerical integration of the differential equations [64].

### 2.2. Simulations of the Electrical Activity of Human Sinoatrial Node Pacemaker Cells

The electrical activity of a single human SAN pacemaker cell was simulated using the comprehensive model of such a cell developed by Fabbri et al. [62], known as the Fabbri–Severi model, with updated equations for the slow delayed rectifier potassium current (I_Ks_) [65]. Vagal tone was simulated by setting the model concentration of acetylcholine (ACh) to 20 nmol/L, whereas β-adrenergic tone was simulated by adopting our ‘High Iso’ settings [65] that are intermediate between the model settings used by Fabbri et al. [62] to simulate the administration of 1 µmol/L of isoprenaline and the model settings that they used to arrive at a pacemaking rate near 180 beats/min.

The CellML code [66] of the Fabbri–Severi model, available from the CellML Model Repository [67] at https://www.cellml.org/ (accessed on 9 June 2023), was edited and run in version 0.9.31.1409 of the Windows-based Cellular Open Resource (COR) environment [68]. All simulations were run for a period of 100 s, which was long enough to achieve stable behavior. The data analyzed are from the final five seconds of this 100 s period.

## 3. Results

First, we reviewed the scientific literature for publications on sinus bradycardia due to a loss-of-function mutation in *HCN4*, in which the clinical data on heart rate were preferably accompanied by in vitro data on the functional effects of the mutation of interest on the *HCN4* current. If the clinical data were not accompanied by such experimental data, we reviewed the scientific literature for separate studies on the functional effects of the specific mutation, resulting in a set of loss-of-function mutations in *HCN4* for which both clinical and in vitro data were available. These mutations are summarized in Figure 1 and include the R375C [69], R378C [70], A414G [71], G480R [72], Y481H [71,73], G482R [71,74,75], A485V [76], K530N [77], R550C [78], R666Q [79], and S672R [80] missense mutations, and the 695X truncating mutation [81].

The clinical and in vitro data resulting from our review are presented in Section 3.1 and Section 3.2, respectively. In these sections, we have only included mutations with quantitative data on heart rate in at least two mutation carriers, thus, for example, ignoring the initial studies by Schulze-Bahr et al. [82] and Ueda et al. [83] on the 573X and D553N mutations, respectively, and the parts of the studies by Schweizer et al. [74] and Möller et al. [70] dealing with the P883R mutation and the R550H and E1193Q mutations, respectively.

In Section 3.3, we use the in vitro data in Section 3.2 to compute the charge carried by I_f_ (Q_f_) during the diastolic depolarization of a prerecorded human SAN AP waveform for each of the collected mutations. In Section 3.4 and Section 3.5, we demonstrate the extent to which Q_f_ can predict the beating rate of a single human sinus node pacemaker cell and the heart rate of mutation carriers, respectively.

### 3.1. Loss-of-Function Mutations in *HCN4*: Clinical Observations

Clinical data on heart rate in the presence of (all heterozygous) loss-of-function mutations in *HCN4* are listed in Table 1 and Table 2, ordered by the location of the mutation sites on the *HCN4* protein (Figure 1). In some studies, data are reported as minimum, average, and maximum heart rates from 24 h Holter recordings. Data from these studies are presented in Table 1. In a similar number of studies, data are reported as resting heart rates and maximum heart rates during exercise testing. Data from the latter studies are listed in Table 2. In several studies, on the G480R mutation by Nof et al. [72], the G482R mutation by Schweizer et al. [74] and Brunet-Garcia et al. [75], the A485 mutation by Laish-Farkash et al. [76], the K530N mutation by Duhme et al. [77], and the 695X mutation by Schweizer et al. [81], both types of data were reported. This explains why these studies appear in both Table 1 and Table 2.

In Table 2, we have omitted the heart rate during exercise testing reported by Alonso-Fernández-Gatta et al. [69] for carriers of the R375C mutation. This is because this heart rate of only 81.5 ± 2.8 beats/min (*n* = 11) was not obtained after completion of the full Bruce protocol for exercise testing but rather after completion of the third stage of this six-stage protocol.

The R375C, G482R, and A485V mutations were also related to sinus bradycardia (and ventricular non-compaction) by Chanavat et al. [84], but these were all single cases, and no quantitative data, either in vivo or in vitro, were provided. In a study of left ventricular non-compaction, Richard et al. [85] presented a family with four heterozygous carriers of the G480C (p.Gly480Cys) mutation, all of whom had sinus bradycardia. However, no quantitative data were provided, neither on the bradycardia itself nor the functional effects of the mutation. The G480C mutation was also identified by Cambon-Viala et al. [86] in a single patient out of a group of 19 *HCN4* mutation carriers with heart rates between 41 and 50 beats/min. Unfortunately, no further clinical data were reported and no functional studies were performed.

### 3.2. Loss-of-Function Mutations in *HCN4*: In Vitro Data

In this section, we present the functional effects of the loss-of-function mutations in *HCN4* collected in Section 3.1, focusing on the functional differences between wild type (WT) and heteromeric mutant *HCN4* channels, again ordered by the location of the mutation sites on the *HCN4* protein (Figure 1). Also, we explain how we translated these experimentally observed differences in *HCN4* current characteristics into changes in the parameter settings of I_f_ in our reconstructions of I_f_ during diastolic depolarization (Section 3.3) as well as in our simulations with the Fabbri–Severi model of a human SAN pacemaker cell [62] (Section 3.4). These changes in parameter settings are summarized in Table 3. The changes are limited to a decrease in the fully activated conductance of I_f_ (g_f_), representing an experimentally observed decrease in the fully activated *HCN4* current, and/or a hyperpolarizing shift in the voltage dependence of the steady-state activation (y_∞_) curve and the bell-shaped (de)activation time constant (τ_y_) curve. The shape of these curves and the effects of hyperpolarizing shifts are illustrated for the R375C mutation in Section 3.2.1 below.

Only a few of the functional studies presented data on the sensitivity of heteromeric *HCN4* mutant channels to cAMP. Such data are limited to the K530N [77] and 695X [81] mutations and are described in the corresponding subsections below. In other cases, data on the sensitivity to cAMP were only gathered for homomeric mutant channels or attempts to obtain data on sensitivity to cAMP failed. For the sake of completeness, these cases are also briefly documented below.

#### 3.2.1. R375C (p.Arg375Cys)

In their recent study of the R375C mutation in *HCN4*, Alonso-Fernández-Gatta et al. [69] carried out whole cell patch clamp experiments at room temperature on Chinese hamster ovary (CHO) cells expressing WT, homomeric R375C mutant, or heteromeric R375C mutant (WT + R375C) *HCN4* channels. Estimated from the graphical representation of their patch clamp data, the fully activated WT + R375C *HCN4* current density at −140 mV was ≈50% of the current density of the WT current, whereas the steady-state activation curve was shifted by ≈−14 mV, as illustrated in Figure 2A, with no change in the reversal potential of the *HCN4* current. The mutation-induced ≈50% decrease in fully activated *HCN4* current was incorporated into the model as a 50% decrease in fully activated conductance of I_f_ (g_f_), whereas the shift in the steady-state activation curve was incorporated as a −14 mV shift in the voltage dependence of the I_f_ activation gate, thus also applying this hyperpolarizing shift to the bell-shaped (de)activation time constant curve, as illustrated in Figure 2B. This latter shift results in an increase in the time constant of activation at highly negative membrane potentials, which would explain the experimentally observed mutation-induced slowing of the *HCN4* current activation at −130 mV.

#### 3.2.2. R378C (p.Arg378Cys)

Möller et al. [70] recorded whole cell currents from *Xenopus* oocytes at room temperature in their study of the R378C mutation. These oocytes were injected with cRNAs to make them express WT, R378C, or WT + R378C *HCN4* channels. Somewhat similar to R375 (Section 3.2.1), the fully activated WT + R378C *HCN4* current was reduced by 57%, the steady-state activation curve was shifted by −7.9 mV, and activation was slowed at −140 mV, which were incorporated into the Fabbri–Severi model as a 57% decrease in the fully activated conductance of I_f_ and a −7.9 mV shift in the voltage dependence of the I_f_ activation gate (Table 3). Consistent with the associated −7.9 mV shift in the bell-shaped time constant curve, Möller et al. [70] observed a decrease in the deactivation time constant of the *HCN4* current at +20 mV. Möller et al. [70] did not observe any ‘dramatic impairment of cAMP activation’ of the mutant channels.

#### 3.2.3. A414G (p.Ala414Gly)

We have recently extended the voltage clamp experiments presented by Milano et al. [71] on CHO cells expressing WT or WT + A414G heterozygous mutant *HCN4* channels, which were also carried out in our laboratory. For details on materials and methods, we, therefore, refer to the study by Milano et al. [71]. Using the amphotericin-perforated patch clamp technique at 36 ± 0.2 °C, we found that the half-maximal activation voltage (V_½_) of the WT + A414G current showed a shift of −19.9 mV relative to WT and the voltage dependence of its (de)activation time constant showed a shift of −11.9 mV, whereas no differences were observed in the slope factor (k) of the steady-state activation curve, the fully activated current density, and the reversal potential. Accordingly, a −19.9 mV shift in the steady-state activation curve and a −11.9 mV shift in the voltage dependence of the time constant of (de)activation were applied to the model I_f_.

#### 3.2.4. G480R (p.Gly480Arg)

In their study of the G480R mutation, Nof et al. [72] injected *Xenopus* oocytes with mRNAs to make them express WT, G480R, or WT + G480R *HCN4* channels. In whole cell voltage clamp experiments at room temperature, they observed that the WT + G480R current activated more slowly and at more negative potentials than the WT current, with no change in the reversal potential. Also, the fully activated WT + G480R current density was substantially smaller than the WT current density. Unfortunately, their data on the heterozygous mutant current are limited. Roughly estimated, the WT + G480R kinetics showed a −10 mV shift in voltage dependence relative to WT, whereas the fully activated current was reduced by 54%. These estimated changes were applied to the I_f_ of the model cell as a combined −10 mV shift of the steady-state activation curve and the bell-shaped time constant curve and a 54% reduction in the fully activated I_f_ conductance. Nof et al. [72] were unable to test the β-adrenergic regulation of either WT or mutant channels in response to epinephrine. It is likely, as suggested by Nof et al. [72], that their attempts failed due to the high levels of endogenous cAMP in their oocytes.

#### 3.2.5. Y481H (p.Tyr481His)

In the study by Milano et al. [71], *HCN4* channels were expressed in CHO cells, and *HCN4* currents were recorded at 37 ± 0.2 °C using the amphotericin-perforated patch clamp technique. The steady-state activation curve of the WT + Y481H mutant current showed a hyperpolarizing shift of as much as 44 mV compared to WT, whereas no difference in its slope factor k was observed. The apparent decrease in fully activated current density at −160 mV did not reach statistical significance. The effects of the mutation were implemented by a −44 mV shift in the steady-state activation curve of I_f_. Given the common observation that a mutation-induced hyperpolarizing shift in the steady-state activation curve of I_f_ is accompanied by a similar shift in the bell-shaped time constant curve, we also applied this shift to the latter curve. Of note, the more recent study by Vermeer et al. [73], which identified a novel family with the Y481H mutation, focused on the structural effects of the mutation and did not include additional patch clamp data.

#### 3.2.6. G482R (p.Gly482Arg)

Clinical data on the G482R mutation were presented by Milano et al. [71], Schweizer et al. [74], Brunet-Garcia et al. [75], and, with a strict focus on ventricular non-compaction, Cambon-Viala et al. [86]. Patch clamp data were only presented by Milano et al. [71] and Schweizer et al. [74]. Using the same experimental approach as for the Y481H mutation (see Section 3.2.5), Milano et al. [71] observed a −39 mV shift in the steady-state activation curve of the WT + G482R mutant current compared to WT, without a change in k. As also observed for the Y481H mutation, there was an apparent decrease in fully activated current density at −160 mV that did not reach statistical significance. In contrast to the findings by Milano et al. [71], Schweizer et al. [74], who expressed *HCN4* channels in human embryonic kidney (HEK-293) cells for their whole cell patch clamp recordings at room temperature, did not observe any change in the steady-state activation curve or kinetic properties of the WT + G482R mutant current compared to WT, whereas the fully activated current density at −120 mV was reduced by 65%. We incorporated the experimental findings of Milano et al. [71] into the Fabbri–Severi model as a −39 mV shift in the steady-state activation curve of I_f_ (applying the same shift to the bell-shaped time constant curve) and those in Schweizer et al. [74] as a 65% decrease in its fully activated conductance.

#### 3.2.7. A485V (p.Ala485Val)

Laish-Farkash et al. [76] injected *Xenopus* oocytes with WT and/or A485V mutant mRNA to let them express WT, A485V, or WT + A485V *HCN4* channels. In whole cell voltage clamp experiments at room temperature, the WT + A485V *HCN4* current activated more slowly and at more hyperpolarized potentials (below −80 vs. −65 mV) than the WT current, with no significant difference in the reversal potential. Also, the fully activated WT + A485V current was substantially smaller than the WT current. As a rough estimate, the *HCN4* current kinetics showed a −15 mV shift in their voltage dependence, whereas the fully activated conductance was reduced by 68%. These estimated changes were applied to I_f_ of the model cell.

#### 3.2.8. K530N (p.Lys530Asn)

In their study of the K530N mutation, Duhme et al. [77] performed whole cell patch clamp experiments at room temperature on transfected HEK-293 cells. Compared to WT, the heteromeric WT + K530N current activated more slowly at −120 mV and showed a −14 mV shift in V_½_, without a change in k, reversal potential, and fully activated current density at −120 mV. These experimental findings were incorporated into the cell model as a −14 mV shift in the voltage dependence of the I_f_ kinetics. Interestingly, the heteromeric WT + K530N channels showed a significantly higher sensitivity to cAMP, with a +7.5 mV larger cAMP-induced depolarizing shift in the steady-state activation curve and a significantly more accelerated activation at −120 mV. Surprisingly, the electrophysiological properties of the homomeric K530N channels were almost indistinguishable from WT channels. In our simulations, we accounted for the larger cAMP-induced shift through a +5 mV larger shift in our ‘High Iso’ settings of the voltage dependence of the I_f_ kinetics, which also resulted in a more accelerated activation at −120 mV.

#### 3.2.9. R550C (p.Arg550Cys)

Campostrini et al. [78] transfected both CHO cells and neonatal rat ventricular cardiomyocytes (NRVCs) with WT, R550C, or WT + R550C h*HCN4* and carried out whole cell patch clamp experiments at room temperature and 36 ± 1 °C, respectively, to assess the functional effects of the R550C mutation. The V_½_ of the WT + R550C current in CHO cells showed a small but statistically significant shift of −4.6 mV compared to WT. No differences were observed in other electrophysiological properties, including k, current density, and time constants of activation and deactivation. Highly similar results were obtained in NRVCs, with a small but statistically significant shift of −3.7 mV in V_½_ and no significant differences in k, current density, and time constants of activation and deactivation at multiple membrane potentials, except for a significantly smaller time constant of deactivation at −65 mV. The effects of the mutation were implemented by a −4 mV shift in the voltage dependence of the I_f_ kinetics, keeping in mind that a −4 mV shift in the membrane potential sensitivity of the time constant of (de)activation would be barely detectable. Homomeric WT and R550C channels showed a very similar sensitivity to cAMP.

#### 3.2.10. R666Q (p.Arg666Gln)

Recently, Wang et al. [79] carried out whole cell patch clamp experiments at room temperature on HEK-293T cells expressing WT, homomeric R666Q mutant, or heteromeric R666Q mutant *HCN4* channels. The V_½_ and k of the WT and mutant *HCN4* currents were highly similar. However, the current density of WT + R666Q at −130 mV was significantly lower than WT. In 24 to 36 h and 36 to 48 h after transfection, it was 50% and 42% of WT, respectively. We incorporated these experimental results into the Fabbri–Severi model as a 54% decrease in the fully activated conductance of I_f_. The sensitivity of homomeric WT and R666Q channels to cAMP was highly similar.

#### 3.2.11. S672R (p.Ser672Arg)

The effects of the S672R mutation were determined by Milanesi et al. [80] through whole cell patch clamp experiments at room temperature on HEK-293 cells that were transfected to express WT or S672R homomeric or heteromeric mutant *HCN4* channels. For heteromeric mutant channels, the changes in electrophysiological properties relative to WT were limited to a −4.9 mV shift in V_½_ and a slight decrease in the time constants of deactivation at multiple membrane potentials. These experimental findings were incorporated into the cell model as a −4.9 mV shift in the voltage dependence of the I_f_ kinetics (i.e., a combined −4.9 mV shift in the steady-state activation curve and the bell-shaped time constant curve). Homomeric WT and S672R channels showed a highly similar sensitivity to cAMP.

#### 3.2.12. 695X (p.695X)

In their study of the 695X mutation, Schweizer et al. [81] performed whole cell patch clamp experiments at room temperature on transfected HEK-293 cells. Under cAMP-free conditions, the V_½_, k, and time constant of activation at −120 mV of the heteromeric WT + 695X channels were highly similar to the WT channels. Changes in electrophysiological properties were observed only in the presence of cAMP (10 µmol/L), as a result of the truncation of the cyclic nucleotide-binding domain (CNBD; Figure 1) due to an insertion of 13 nucleotides in exon 6 of the *HCN4* gene that leads to a premature stop codon [81], which exerted a dominant-negative effect on the cAMP-induced increase in *HCN4* current. These experimental findings were incorporated into the cell model as a fixed −10.9 mV shift in the voltage dependence of the I_f_ kinetics (which is the maximum shift at high ACh concentrations in the model equations).

### 3.3. Q_f_: Charge Carried by I_f_ during Diastolic Depolarization

Using the in vitro data on the biophysical effects of the loss-of-function mutations in *HCN4* in Section 3.2, we computed the charge that is carried by the associated heteromeric mutant I_f_ during diastolic depolarization of a human SAN pacemaker cell (Q_f_). To this end, we simulated AP clamp experiments using a prerecorded human SAN AP waveform [61] and I_f_ equations based on our patch clamp data on I_f_ in human SAN cells [63]. As already set out in Section 2.1, these equations are also part of the Fabbri–Severi model [62].

Figure 3, A and B, show the prerecorded AP with its diastolic depolarization and the associated reconstructed WT I_f_, which contributes to the diastolic depolarization as an inward current that carries a charge of 1.00 pC (Figure 3B, filled area). Reconstructing I_f_ with its g_f_ halved and its voltage dependence shifted by −14 mV, as listed in Table 3 and set out in Section 3.2.1, we obtained the WT + R375C I_f_, which is much smaller in amplitude and carries a charge of only 0.20 pC during diastolic depolarization (Figure 3C). Similarly, we reconstructed I_f_ for each of the other (heterozygous) mutations listed in Table 3 (and described in detail in Section 3.2) and computed Q_f_. The obtained data on Q_f_ are summarized in Table 4.

### 3.4. Can Q_f_ Predict the Beating Rate of a Single Human Sinus Node Pacemaker Cell?

We questioned to what extent the data on Q_f_ obtained from (simulated) AP clamp experiments can predict the beating rate of a single human sinus node pacemaker cell. Therefore, we performed computer simulations with the Fabbri–Severi model under control conditions (default ‘wild-type’ I_f_) and for each of the (heterozygous) mutations listed in Table 3 at different levels of autonomic tone. Figure 4 shows the results that we obtained for the WT + R375C mutation compared to WT. Under vagal tone (20 nmol/L ACh; Figure 4A), the modulatory I_f_ is already small. Yet, the cycle length is substantially increased due to the mutation-induced decrease in I_f_, and the beating rate is reduced from 42.4 to 29.3 beats/min. With the default model (no rate modulation; Figure 4B), where I_f_ is larger than under vagal tone, the mutation-induced decrease in I_f_ results in a decrease in the beating rate from 70.2 to 56.9 beats/min. Under β-adrenergic tone (‘High Iso’; Figure 4C), I_f_ exerts its modulatory role and is again larger. The mutation-induced decrease in I_f_ now results in a decrease in the beating rate from 109.5 to 81.2 beats/min.

Having obtained data on the beating rate, as illustrated in Figure 4, for each of the mutations, we plotted these beating rates against the associated Q_f_ from the simulated AP clamp experiments (Table 4), resulting in Figure 5. At each level of autonomic tone, there is a very strong correlation between the beating rate and Q_f_, suggesting that data on Q_f_ obtained in AP clamp experiments on cells expressing the (heteromeric) *HCN4* channels of interest, compared to Q_f_ for wild type channels, may predict the amount of sinus bradycardia in mutation carriers.

### 3.5. Can Q_f_ Predict the Heart Rate of Mutation Carriers?

Keeping in mind that the promising results of Section 3.4 are based on the ideal case of an I_f_ in a (simulated) human SAN cell with biophysical parameters that are completely identical to those of the *HCN4* current in an expression system, we assessed to what extent the data on Q_f_ obtained from (simulated) AP clamp experiments can predict the clinically observed heart rates of mutation carriers. To this end, we first plotted the minimum, average, and maximum heart rates from 24 h Holter recordings (Table 1) against the associated Q_f_ (Table 4), resulting in Figure 6. The clinically observed minimum heart rate shows a strong correlation with Q_f_ (R^2^ = 0.73; *p* < 0.001, ANOVA). The average heart rate shows a less clear relationship (R^2^ = 0.56) but is still statistically significant (*p* = 0.002). For the maximum heart rate, the relationship is not statistically significant (*p* = 0.07).

We also plotted the clinically observed resting heart rates and maximum heart rates during exercise testing in Table 2 against the associated Q_f_. As shown in Figure 7A, the resting heart rate shows a strong correlation with Q_f_ (R^2^ = 0.71; *p* = 0.001). Data on the maximum heart rate during exercise testing are limited, and no clear correlation can be discerned (Figure 7B).

## 4. Discussion

These days, risk stratification of ion channelopathies is largely based on changes in the electrophysiological properties of ion channel variants. So far, such risk stratification has been performed not only for a selection of cardiac ion channelopathies, as already outlined in the Introduction section, but also for neuromuscular diseases with Na_V_1.4 variants [87], hearing loss related to *KCNQ4* variants [88], and encephalopathies, including schizophrenia, with *CACNA1I* variants [89], and epilepsy with *SCN1A* [90], *KCNB1* [91], *HCN1* [92], *KCNQ2* [93], and *SCN2A* [94] variants. In the present study, we tested whether risk stratification for sinus bradycardia can be based on AP clamp experiments on transfected cells to compute the charge carried by (mutant) I_f_ during the diastolic depolarization phase of a prerecorded human SAN AP, using this AP as command potential. The mutation-induced changes in the biophysical parameters of an *HCN4* variant of interest are summarized in the mutation-induced change in this charge, without the need to characterize each of these parameters separately.

In our study, we first reviewed the scientific literature for publications on loss-of-function mutations in *HCN4* with both clinical and in vitro data on the effects of the mutation. This resulted in clinical data on heart rate (Table 1 and Table 2) and associated in vitro data on (changes in) *HCN4* current characteristics (Table 3) for a total of 12 mutations (Figure 1). This rather small number of mutations highlights the unfortunate situation that clinical studies identifying mutations in *HCN4* with potentially bradycardic effects are not always accompanied by in vitro data on their functional effects. Perhaps the best example is the extensive clinical study by Hategan et al. [95], which identified the novel c.1737 + 1 G > T splice-site mutation in *HCN4* in a large family with familial bradycardia. Although it is highly likely that the c.1737 + 1 G > T mutation is disease-causing, it would be very interesting to know its functional effects. This is all the more important for less extensive clinical studies, where only one or a few mutation carriers have been identified. The importance of in vitro data in such studies is, for example, underscored in the study of Erlenhardt et al. [96], who identified the V759I mutation in *HCN4*, which had previously been identified in cases of sudden infant death syndrome [97] and sudden unexpected death in epilepsy in a patient with severe sinus bradycardia [98]. Erlenhardt et al. [96] were the first to perform functional studies on the mutation they identified. Patch clamp experiments on *Xenopus* oocytes showed that voltage dependence, activation kinetics, sensitivity to cAMP, and cell surface expression of mutant channels were all indistinguishable from wild type channels.

After obtaining our clinical and in vitro data, we used the in vitro data to compute Q_f_—i.e., the charge carried by I_f_ during the diastolic depolarization phase of a prerecorded human sinus node action potential waveform (Figure 3)—in simulated AP clamp experiments for each of the mutations as a potential measure of their severity. Also, we used these in vitro data to test the functional effects of each of the mutations on the beating rate of the comprehensive Fabbri–Severi model of a human sinus node cell (Figure 4). These in silico experiments revealed a very strong correlation between the beating rate and Q_f_ (Figure 5), suggesting that this (relatively) readily obtained Q_f_ may prove a useful tool for risk stratification of sinus bradycardia due to loss-of-function mutations in *HCN4*. A high correlation could be anticipated from the use of identical I_f_ equations [63] in our reconstructions of I_f_ under action potential clamp conditions and in the Fabbri–Severi model [62].

With R^2^ values of 0.73 and 0.71, respectively, the clinically observed minimum or resting heart rates (Table 1 and Table 2) show a strong correlation with Q_f_ (Figure 6 and Figure 7). However, the maximum heart rate during 24 h Holter recordings (Figure 6) and during exercise testing (Figure 7) do not show a clear correlation with Q_f_. This may reflect a less pronounced role of I_f_ under β-adrenergic tone, when other ion currents contributing to diastolic depolarization are substantially upregulated, in particular the L-type and T-type calcium currents [99,100]. Alternatively, or simultaneously, the loss of a clear correlation between the clinical data and Q_f_ at high rates may point to shortcomings in the ‘High Iso’ settings that we applied to the Fabbri–Severi model to simulate β-adrenergic tone [65]. One such shortcoming is that the T-type calcium current is not considered a β-adrenergic target. Fabbri et al. [62] had to develop their model in the absence of an electrophysiological characterization of the β-adrenergic targets in the human sinus node, thus potentially requiring updates when quantitative data from human tissue become available. Furthermore, it should be kept in mind that the Fabbri–Severi model is a single-cell model, whereas the interaction of the human sinus node with its atrial surroundings is also dependent on β-adrenergic tone, as individual ion currents of the atrial myocytes are also up- or downregulated under β-adrenergic tone [101,102].

In the present study, we based our computations of Q_f_ on the in vitro data that are summarized in Table 3. Thus, we implicitly assumed that all of the experimentally observed mutation-induced effects on *HCN4* currents in expression systems also apply to I_f_ in human SAN pacemaker cells from mutation carriers under physiological conditions. Although all mutation-induced effects listed in Table 3 are relative changes (i.e., reductions in fully activated conductance and/or shifts in the voltage dependence of steady-state activation and time constants of (de)activation), it may well be that these mutation-induced effects are quantitatively or even qualitatively dependent on the specific expression system and recording temperature or other recording conditions, which show essential differences between studies from different laboratories (Table 3). In a direct comparison of the effects of the p.T1620M mutation in the *SCN5A* gene, encoding the pore-forming α-subunit of the cardiac Na_V_1.5 fast sodium channel, Baroudi et al. [103] even obtained opposite phenotypes depending on the expression system (*Xenopus* oocytes vs. mammalian tsA201 cells; both at room temperature). Studying the same p.T1620M mutation in *SCN5A* in tsA201 cells at 22 °C and 32 °C, Dumaine et al. [104] observed a mutation-induced acceleration of current decay at 32 °C but not 22 °C, directly demonstrating a temperature dependence of the nature of the mutation-induced effects. Differences in the expression system (CHO vs. HEK-293 cells) and recording conditions (37 ± 0.2 °C vs. room temperature; amphotericin-perforated vs. whole cell patch clamp technique) may explain, at least in part, why Milano et al. [71] and Schweizer et al. [74] obtained essentially different data on WT + G482R channels (−39 mV shift in the steady-state activation curve vs. 65% reduction in fully activated conductance). In contrast, Campostrini et al. [78] obtained very similar data on WT + R550C channels in CHO cells at room temperature and in NRVCs at 36 ± 1 °C (both whole cell patch clamp recordings).

## 5. Conclusions

While a translational perspective remains to be seen, we conclude that AP clamp on transfected cells, without the need for further voltage clamp experiments and data analysis to determine individual biophysical parameters of I_f_, is a promising tool for risk stratification of sinus bradycardia due to loss-of-function mutations in *HCN4*. In combination with an I_f_ blocker, this tool may also prove useful when applied to human-induced pluripotent stem cell-derived cardiomyocytes (hiPSC-CMs) obtained from mutation carriers and non-carriers.

## Figures and Tables

**Figure 1 biomedicines-11-02447-f001:**
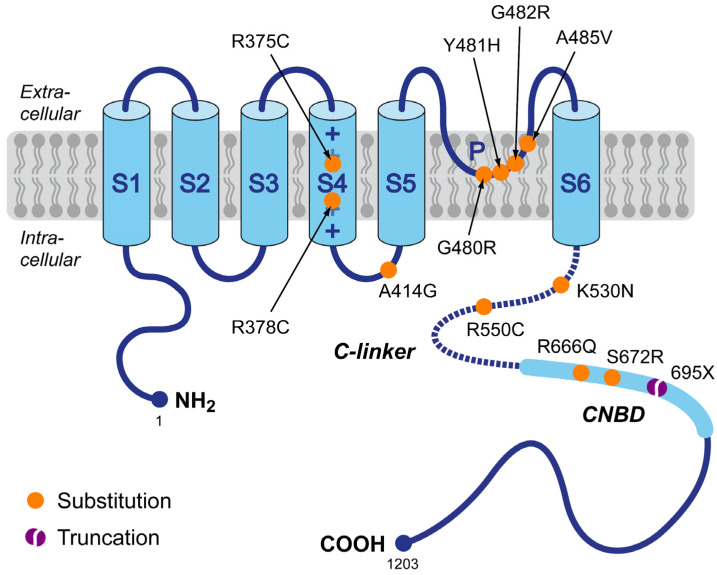
Schematic topology of the *HCN4* protein and the set of loss-of-function mutations in *HCN4* associated with familial sinus bradycardia for which both clinical and in vitro data were available, requiring that these clinical data include quantitative heart rate data from at least two mutation carriers. Tetramers of *HCN4* α-subunits form the cardiac ion channels that conduct the hyperpolarization-activated ‘funny’ current (I_f_). The *HCN4* protein has six transmembrane segments (S1–S6), a pore-forming loop (P), and intracellular N- and C-termini. The voltage sensor of the channel is formed by the positively charged S4 helix. The C-terminus contains the C-linker (dotted line) and the cyclic nucleotide-binding domain (CNBD), which is known to mediate cyclic AMP (cAMP)-dependent changes in HCN channel gating. Colored dots indicate the location of the loss-of-function mutations in the *HCN4* protein of the present study. This set of mutations includes eleven substitutions (R375C, R378C, A414G, G480R, Y481H, G482R, A485V, K530N, R550C, R666Q, and S672R) and one truncation (695X).

**Figure 2 biomedicines-11-02447-f002:**
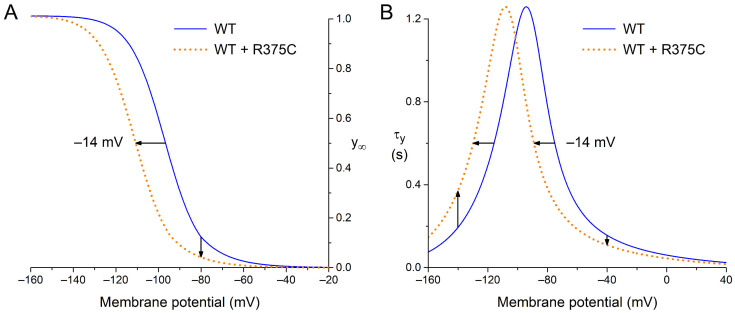
Voltage dependence of wild type (WT; solid blue lines) and heteromeric R375C mutant (WT + R375C; orange dotted lines) *HCN4* current (de)activation. (**A**) Steady-state activation (y_∞_). Horizontal arrow: the mutation-induced hyperpolarizing shift in half-maximum activation voltage. Vertical arrow: the mutation-induced decrease in maximally available *HCN4* current. (**B**) Time constant of (de)activation (τ_y_). Horizontal arrow: the mutation-induced hyperpolarizing shift in the bell-shaped curve. Upward vertical arrow: the mutation-induced decrease in the rate of (de)activation at highly negative membrane potentials. Downward vertical arrow: the mutation-induced increase in the rate of (de)activation at less negative membrane potentials.

**Figure 3 biomedicines-11-02447-f003:**
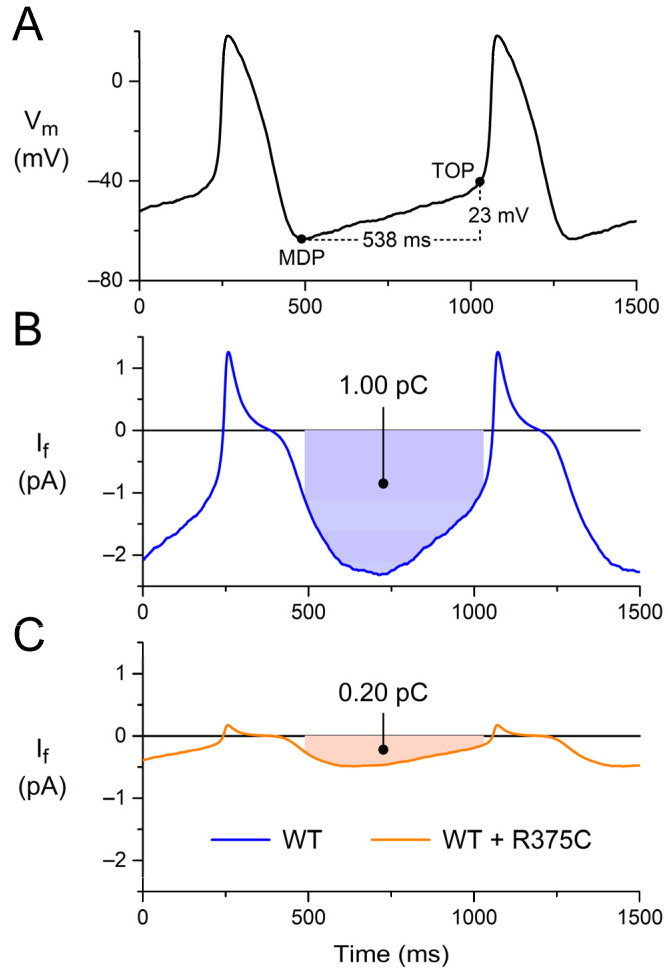
Charge carried by I_f_ during diastolic depolarization. (**A**) Prerecorded AP waveform of an isolated human sinus node pacemaker cell. During the diastolic depolarization from the maximum diastolic potential (MDP) to the take-off potential (TOP), which takes 538 ms, the membrane potential (V_m_) depolarizes by 23 mV. (**B**) Associated reconstructed WT I_f_, which carries a charge of 1.00 pC (filled area) as an inward current during diastolic depolarization. (**C**) Associated reconstructed WT + R375C I_f_, which carries a charge of 0.20 pC during diastolic depolarization. The AP waveform of panel A is a typical waveform obtained from a set of single isolated human SAN pacemaker cells [61], and the I_f_ curve of panel B is reconstructed from this typical AP waveform and the I_f_ equations of the Fabbri–Severi model [62], which are based on the I_f_ data obtained in voltage clamp experiments on the same set of single-isolated human SAN pacemaker cells [61,63].

**Figure 4 biomedicines-11-02447-f004:**
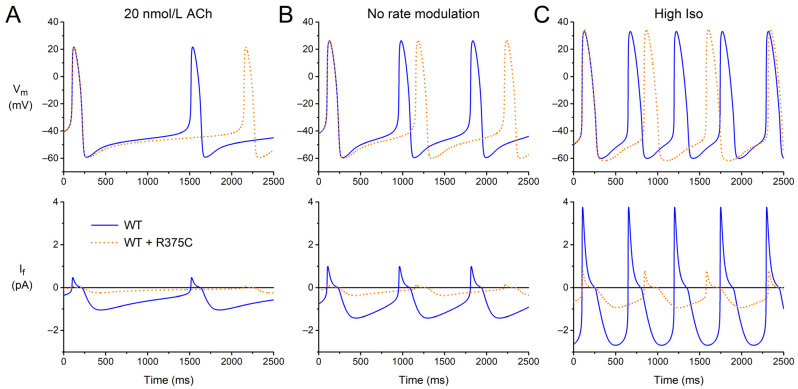
Electrical activity of the Fabbri–Severi model of a human SAN pacemaker cell with its default ‘wild-type’ I_f_ (WT; solid blue lines) and heteromeric R375C mutant I_f_ (WT + R375C; orange dotted lines) at different levels of autonomic tone. (**A**) Vagal tone (simulated ACh concentration of 20 nmol/L). (**B**) No rate modulation (default model). (**C**) β-Adrenergic tone (‘High Iso’ settings of the model).

**Figure 5 biomedicines-11-02447-f005:**
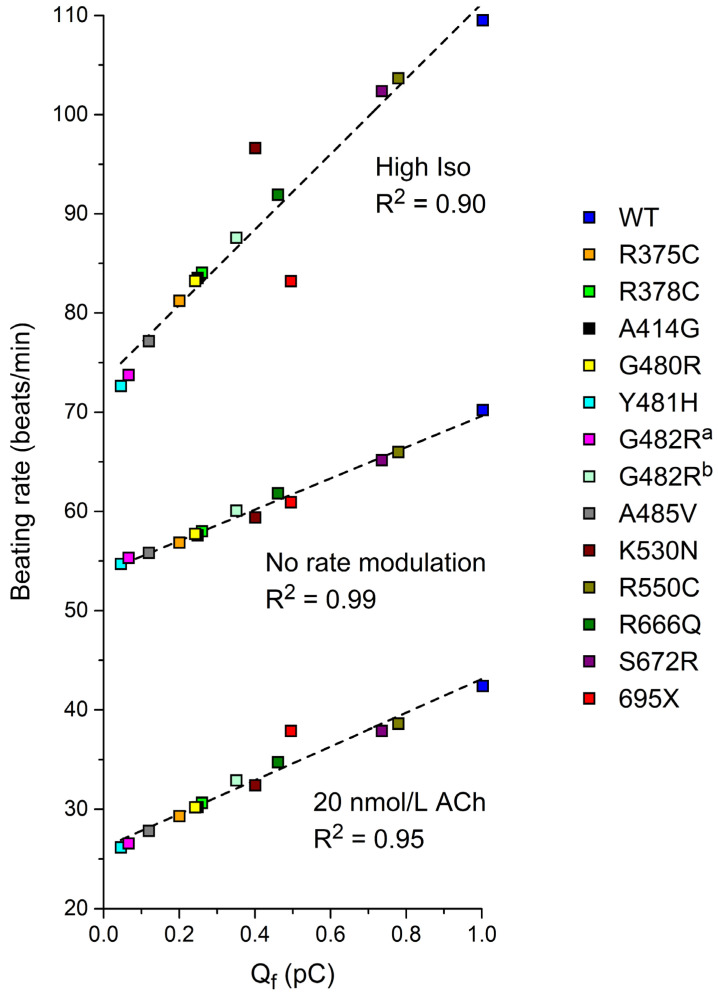
The beating rate of the Fabbri–Severi model of a human SAN pacemaker cell with its default ‘wild-type’ I_f_ (WT) and heteromeric mutant I_f_, simulated with the settings presented in Section 3.2 as a function of Q_f_ (Section 3.3) at different levels of autonomic tone. ^a^ I_f_ parameters are based on Milano et al. [71]. ^b^ I_f_ parameters are based on Schweizer et al. [74]. Dashed lines are linear fits.

**Figure 6 biomedicines-11-02447-f006:**
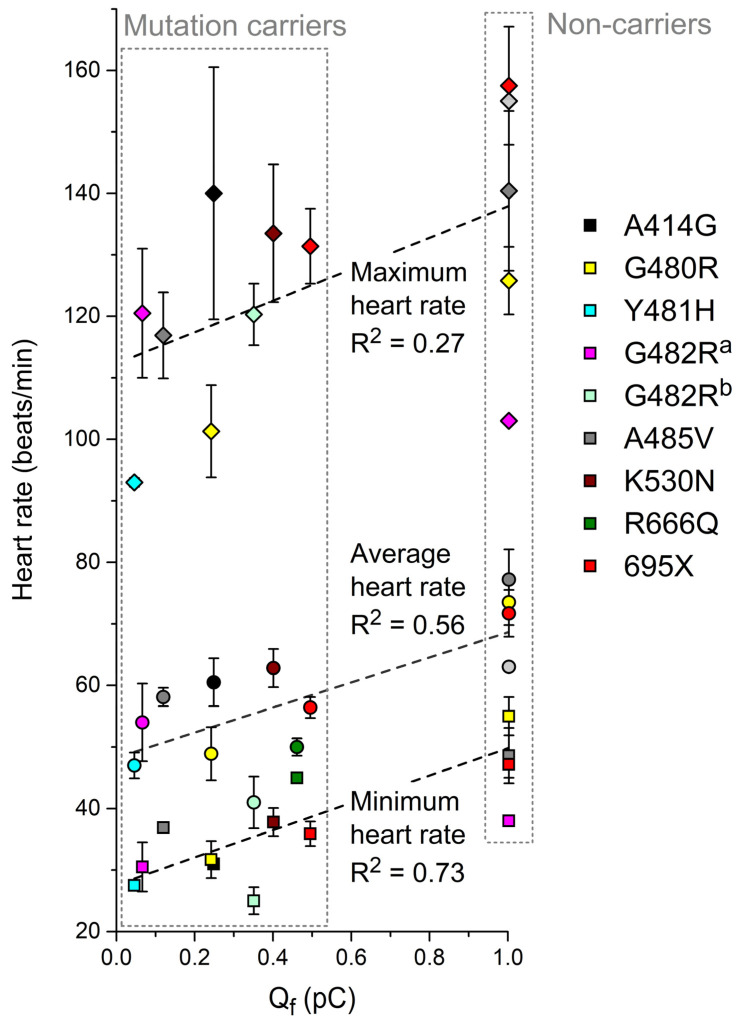
Minimum, average, and maximum heart rates obtained during 24 h Holter recordings from heterozygous carriers of the mutations in *HCN4* as indicated or from non-carriers of the same family (Table 1) as a function of Q_f_ (Section 3.3). ^a^ I_f_ parameters are based on Milano et al. [71]. ^b^ I_f_ parameters are based on Schweizer et al. [74]. Dashed lines are linear fits.

**Figure 7 biomedicines-11-02447-f007:**
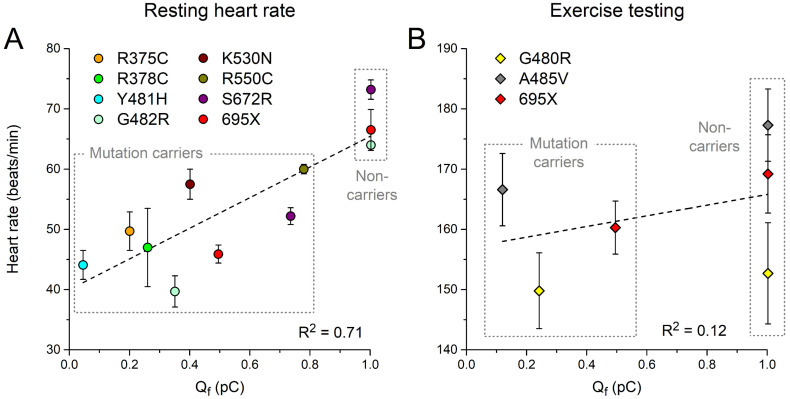
(**A**) Resting heart rates and (**B**) maximum heart rates during exercise testing from heterozygous carriers of the mutations in *HCN4* as indicated or from non-carriers of the same family (Table 2) as a function of Q_f_ (Section 3.3). Dashed lines are linear fits.

**Table 1 biomedicines-11-02447-t001:** Minimum, average, and maximum heart rates from 24 h Holter recordings.

Mutation	Group	Heart Rate (Beats/Min) *			Reference
		Minimum	Average	Maximum	
A414G	Carriers (*n* = 2)	31.0 ± 0.7	60.5 ± 3.9	140.0 ± 20.5	Milano et al. [71]
G480R	Carriers (*n* = 7)	31.7 ± 3.0	48.9 ± 4.3	101.3 ± 7.5	Nof et al. [72]
	Non-carriers (*n* = 8)	55.0 ± 3.1	73.5 ± 3.7	125.8 ± 5.5	
Y481H	Carriers (*n* = 2)	27.5 ± 0.4	47.0 ± 2.1	93.0 ± 0.7	Milano et al. [71]
G482R	Carriers (*n* = 6)	30.5 ± 4.0	54.0 ± 6.3	120.5 ± 10.5	Milano et al. [71]
	Non-carrier	38	63	103	
G482R	Carriers (*n* = 3)	25.0 ± 2.2	41.0 ± 4.2	120.3 ± 5.0	Schweizer et al. [74]
	Non-carrier	48	63	155	
G482R	Carriers (*n* = 3)	29.3 ± 3.7			Brunet-Garcia et al. [75]
	Carriers (*n* = 2)		55 ± 4.2	159.0 ± 4.2	
A485V	Carriers (*n* = 14)	36.9 ± 0.8	58.1 ± 1.5	116.9 ± 7.0	Laish-Farkash et al. [76]
	Non-carriers (*n* = 5)	48.6 ± 4.5	77.2 ± 4.9	140.4 ± 13.0	
K530N	Carriers (*n* = 6)	37.8 ± 2.3	62.8 ± 3.1	133.5 ± 11.2	Duhme et al. [77]
R666Q	Carriers (*n* = 2)	45.0 ± 0.7	50.0 ± 1.4		Wang et al. [79]
695X	Carriers (*n* = 7)	35.9 ± 2.0	56.4 ± 1.7	131.4 ± 6.1	Schweizer et al. [81]
	Non-carriers (*n* = 6)	47.2 ± 2.2	71.7 ± 3.8	157.5 ± 9.6	

* Data on heart rate are mean ± SEM for mutation carriers and, if available, for non-carriers from the same family.

**Table 2 biomedicines-11-02447-t002:** Resting heart rates and maximum heart rates during exercise testing.

Mutation	Group	Heart Rate (Beats/Min) *		Reference
		Resting	Exercise Testing	
R375C	Carriers (*n* = 12)	49.7 ± 3.2		Alonso-Fernández-Gatta et al. [69]
R378C	Carriers (*n* = 3)	47.0 ± 6.5		Möller et al. [70]
G480R	Carriers (*n* = 6)		149.8 ± 6.3	Nof et al. [72]
	Non-carriers (*n* = 6)		152.7 ± 8.4	
Y481H	Carriers (*n* = 8)	44.1 ± 2.4		Vermeer et al. [73]
G482R	Carriers (*n* = 3)	39.7 ± 2.6		Schweizer et al. [74]
	Non-carrier	64		
G482R	Carriers (*n* = 2)	44.0 ± 4.2		Brunet-Garcia et al. [75]
A485V	Carriers (*n* = 8)		166.6 ± 6.0	Laish-Farkash et al. [76]
	Non-carriers (*n* = 4)		177.3 ± 6.0	
K530N	Carriers (*n* = 4)	57.5 ± 2.5		Duhme et al. [77]
R550C	Carriers (*n* = 3)	60.0 ± 0.8		Campostrini et al. [78]
S672R	Carriers (*n* = 15)	52.2 ± 1.4		Milanesi et al. [80]
	Non-carriers (*n* = 12)	73.2 ± 1.6		
695X	Carriers (*n* = 8)	45.9 ± 1.5		Schweizer et al. [81]
	Carriers (*n* = 7)		160.3 ± 4.4	
	Non-carriers (*n* = 6)	66.5 ± 3.4	169.2 ± 6.5	

* Data on heart rate are mean ± SEM for mutation carriers and, if available, for non-carriers from the same family.

**Table 3 biomedicines-11-02447-t003:** Heteromeric mutation-induced changes in I_f_ characteristics derived from *HCN4*-transfected cells.

Mutation	Expression System	Recording System	Recording Temperature	Scaling Factor for g_f_	Shift in V_m_ Dependence (mV)	Reference
R375C	CHO	Whole cell	Room (21–23 °C)	0.5	−14	Alonso-Fernández-Gatta et al. [69]
R378C	*Xenopus*	Two-electrode	Room (22–24 °C)	0.43	−7.9	Möller et al. [70]
A414G	CHO	Amphotericin-perforated	36 ± 0.2 °C	1	−19.9 (y_∞_), −11.9 (τ_y_)	Milano et al. [71]; present study
G480R	*Xenopus*	Two-electrode	Room (21–23 °C)	0.46	−10	Nof et al. [72]
Y481H	CHO	Amphotericin-perforated	37 ± 0.2 °C	1	−44	Milano et al. [71]
G482R	CHO	Amphotericin-perforated	37 ± 0.2 °C	1	−39	Milano et al. [71]
G482R	HEK-293	Whole cell	Room (21–23 °C)	0.35	0	Schweizer et al. [74]
A485V	*Xenopus*	Two-electrode	Room (21–23 °C)	0.32	−15	Laish-Farkash et al. [76]
K530N	HEK-293	Whole cell	Room (21–23 °C)	1	−14	Duhme et al. [77]
R550C	CHO	Whole cell	Room	1	−4	Campostrini et al. [78]
	NRVC	Whole cell	36 ± 1 °C			
R666Q	HEK-293T	Whole cell	Room (21–23 °C)	0.46	0	Wang et al. [79]
S672R	HEK-293	Whole cell	Room (25–26 °C)	1	−4.9	Milanesi et al. [80]
695X	HEK-293	Whole cell	Room (21–23 °C)	1	−10.9	Schweizer et al. [81]

I_f_: hyperpolarization-activated ‘funny’ current; g_f_: fully activated I_f_ conductance; V_m_: membrane potential; CHO: Chinese hamster ovary cells; HEK-293(T): human embryonic kidney 293(T) cells; NRVC: neonatal rat ventricular cardiomyocytes; *Xenopus*: *Xenopus* oocytes; amphotericin-perforated: amphotericin-perforated patch clamp technique; two-electrode: two-electrode voltage-clamp technique; whole cell: whole cell patch clamp technique; room: room temperature; y_∞_: steady-state activation; τ_y_: (de)activation time constant.

**Table 4 biomedicines-11-02447-t004:** Charge carried by I_f_ during diastolic depolarization of a human SAN pacemaker cell.

Mutation	Charge Carried by I_f_ during Diastolic Depolarization (pC)
Wild type	1.003
R375C	0.201
R378C	0.260
A414G	0.249
G480R	0.242
Y481H	0.046
G482R ^a^	0.066
G482R ^b^	0.351
A485V	0.120
K530N	0.401
R550C	0.780
R666Q	0.461
S672R	0.736
695X	0.495

Data obtained from simulated action potential (AP) clamp experiments. ^a^ I_f_ parameters are based on Milano et al. [71]. ^b^ I_f_ parameters are based on Schweizer et al. [74].

## Data Availability

All data are available from the academic researchers upon request.

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
