# Peer review of "The Action Potential Clamp Technique as a Tool for Risk Stratification of Sinus Bradycardia Due to Loss-of-Function Mutations in HCN4: An In Silico Exploration Based on In Vitro and In Vivo Data"

_biomedicines, 2023, doi:10.3390/biomedicines11092447_

Round 1

Reviewer 1 Report

The manuscript by Verkerk and Wilders entitled “Sinus Bradycardia due to Loss-of-Function Mutations in HCN4: Risk Stratification by Action Potential Clamp” uses a risk classification assay on HCN4 loss-of-function mutations from which both clinical and in vitro experimental data were available from the literature.  The experimental data was then used to compute the charge carried by If (Qf) during the diastolic depolarization phase of a prerecorded human sinus node action potential waveform.  The authors utilize the Fabbri–Severi model of a human SAN pacemaker cell to determine if Qf predicts (1) the beating rate associated with mutation-induced changes in If, and (2) the heart rate observed in patients carrying the associated mutation in HCN4.  The authors suggest that the beating rate of the model cell as well as the clinically observed minimum or resting heart rate show a strong correlation with Qf.

General Comments

I thought the Results of the manuscript was nicely written.  The use of a mathematical model to predict changes in heart associated with loss of function mutations in HCN4 is interesting. However, the Methods section could be improved to enable the reader to understand the nature of the Methods used in the Manuscript (specifics are given below).  I do have some concerns which the authors should consider.    

Major Comments.

1) I feel the Materials and Methods section could use some enhancement.  I have some suggestions that I feel would improve the understanding of the Methods.

 a) How was the SA nodal action potential acquired?  I appreciated the authors reference their previous paper but maybe a few details in the present manuscript would be of benefit to the reader.  Perforated patch vs ruptured patch?  What were internal and external solutions.  Temperature of recordings?

 b) Regarding the data shown in Figure 3A-B, does the data represent actual recordings from a human SAN cell under AP clamp conditions?  The WT information is critical as the effect of the loss-of function mutation data is based off the WT data.  

 2)  The authors do a nice job of assembling functional data from various studies examining HCN4 loss-of function mutations to incorporate into the model.  My concern is the methodologies used by the various labs to acquire the data.  As the authors are aware, data acquired from Xenopus oocytes can be dramatically different compared to data acquired from mammalian cells (Baroudi et al., 2000).  Similarly, room temperature vs. body temperature can also affect results (Dumaine et al., 1999). These parameters can affect the time and voltage dependence of If as well as the physiological role in the regulation of SA node action potential depolarization.  My questions are have the authors considered these differences? Does the model incorporate Q10 changes when functional studies were performed at room temperature?   

 Minor comments

1) Similar to Major Comment 1a, is the maximum current of 2 pA as depicted on the Y-axis of Figure 3B correct? 

2)  The authors show a modest correlation between charge movement and resting heart (R2=0.56 in Figure 6 and R2=0.71 in Figure 7).  However, there was no correlation when heart rate was high (i.e under exercise conditions).  Perhaps the authors can discuss the reason for the differences and speculate on mechanisms.  I would surmise that exercise has increased adrenergic activity which is affecting other ion channels aside from If (HCN4).

Reviewer 2 Report

The aim of the study is to assess whether the patch-clamp analyses performed on heteromeric loss of function mutations of the HCN4 pacemaker channel can provide useful elements for risk stratification of Sick Sinus Syndrome and more specifically of sinus bradycardia. The authors have carried out a systematic review of the literature published on this type of mutations and have gathered clinical (heart rates: minimal, maximal, average, resting, and during exercise of heterozygous carriers) and single cell patch-clamp parameters (maximal conductance and voltage-dependent shift from heteromeric expression). Using the Fabbri-Severi (F-S) model of the SAN action potential, the authors were able to measure for each of the mutation considered the amount of charge (Qf) flowing through heteromeric mutant pacemaker channels during the diastolic depolarization. These Qf values were then put in relation to the beating rate of the Fabbri-Severi model and to the heart rates of the patients. The F-S model was used to estimates action potential rates under normal, ACh, and ISO stimulation, while for the heart rates values the authors used those directly assessed from the patients. The correlation between the Qf values and the F-S model rates was significant at all rates tested (slow/ACh, normal/control, high/ISO); the correlation between the Qf values and clinical heart rates was also significant with the exception of conditions of high (maximal and exercise) heart rates.

The authors conclude that this approach, and particularly the assessment of the Qf parameter, represents a promising tool for risk stratification of the bradycardia associated with HCN4 mutations.

In my opinion the study is well designed. However the concept of stratification remains within basic electrophysiology and modeling studies.

I have few points:

1)      The finding that the AP rates of the F-S model correlate well with the Qf amplitudes is expected since the Qf is indeed estimated using this model. I would actually be surprised if the correlation would be absent.

2)      The authors should consider adding some comments on the reason why the “clinical” correlation is lost at high heart rates. I wonder whether this lack of Qf/high heart rate correlation represents a weakness of (or better a stimulus to implement) the F-S model. Indeed, the model is coherent with itself (point 1) also at high rates, but it is not with clinical data.

3)      A major point is that this “stratification” represents an important tool for cell electrophysiologists since is based on parameters that directly derive from patch-clamp studies. Unfortunately, I do not see a translational perspective. I believe that this concept should appear in the conclusions.

Reviewer 3 Report

This paper addresses an important and interesting topic that action potential clamp on transfected cells, without the need for further voltage clamp experiments and data analysis to determine individual biophysical parameters of If。The authors tested whether risk stratification for sinus bradycardia can be based on AP clamp experiments on transfected cells to compute the charge carried by (mutant) If during the diastolic depolarization phase of a prerecorded human SAN AP, using this AP as command potential. The results show that Qf could predict the beating rate of a single human sinus node pacemaker cell and predict the heart rate of mutation carriers. Overall, the article is well organized and its presentation is good. However, some minor issues still need to be improved:

(1) What are the inclusion criteria for these mutations and the exclusion criteria for other mutations? For example, the R666Q is not included (PMID:36244448).

(2) Are these mutations emission sites? Is there a familial mutation of sick sinus syndrome?

(3) The authors should summarize the main results of this paper of the functional studies presented data on the sensitivity of heteromeric HCN4 mutant channels in result 3.2.1-11.

(4) I suggest that methods part could be clearly introduced and the limitation of this work should be discussed.

The paper was well written and English language fine.
